# High Normal Range of Free Thyroxine is Associated with Decreased Triglycerides and with Increased High-Density Lipoprotein Cholesterol Based on Population Representative Data

**DOI:** 10.3390/jcm8060758

**Published:** 2019-05-28

**Authors:** Jeongmin Lee, Jeonghoon Ha, Kwanhoon Jo, Dong-Jun Lim, Jung-Min Lee, Sang-Ah Chang, Moo-Il Kang, Min-Hee Kim

**Affiliations:** 1Division of Endocrinology and Metabolism, Department of Internal Medicine, Eunpyeong St. Mary’s Hospital, College of Medicine, The Catholic University of Korea, Seoul 03391, Korea; 082mdk45@catholic.ac.kr (J.L.); leejm68@catholic.ac.kr (J.-M.L.); sangah@catholic.ac.kr (S.-A.C.); 2Division of Endocrinology and Metabolism, Department of Internal Medicine, Seoul St. Mary’s Hospital, College of Medicine, The Catholic University of Korea, Seoul 06591, Korea; hajhoon@catholic.ac.kr (J.H.); ldj6026@catholic.ac.kr (D.-J.L.); mikang@catholic.ac.kr (M.-I.K.); 3Division of Endocrinology and Metabolism, Department of Internal Medicine, Incheon St. Mary’s Hospital, College of Medicine, The Catholic University of Korea, Incheon 21431, Korea; lovi@catholic.ac.kr

**Keywords:** euthyroid, thyroid hormone, dyslipidemia

## Abstract

*Background*: We aimed to evaluate the association between thyroid hormone (free thyroxine, free T4) level and lipid profiles in nationally representative data. *Methods*: This study was based on cross-sectional survey data from the sixth Korea National Health and Nutrition Examination Survey IV. After exclusion of subjects with a history of thyroid disease or abnormal thyroid function test and those on medication for dyslipidemia and/or cardiovascular disease, a total of 3548 subjects were included in the study. *Results*: There was a significant decrease in serum triglyceride levels and increase in serum high-density lipoprotein (HDL) cholesterol levels with high free T4 quartiles after adjustment for confounding factors (*p* for trend = 0.001 and *p* for trend = 0.014, respectively). Risk of hypertriglyceridemia was significantly decreased (odds ratio of 0.72 (95% confidential interval 0.53–0.98)) in the highest free T4 quartile compared to the lowest free T4 quartile, *p* = 0.044). *Conclusions*: Serum free T4 levels within normal range negatively correlated with serum triglyceride level and positively correlated with HDL-cholesterol level. Therefore, a close surveillance in terms of lipid profiles could be considered in subjects with low normal serum free T4 levels.

## 1. Introduction

Thyroid hormones play a pivotal role in the regulation of lipid metabolism. They directly regulate hepatic lipid metabolism including de novo lipogenesis, triglyceride (TG) assembly, lipolysis, fatty acid oxidation, and biosynthesis and clearance of cholesterol [1]. They change the composition of lipoproteins by reducing serum low-density lipoprotein (LDL) cholesterol through stimulating LDL-receptor genes [2]. Thyroid hormones also activate the cholesteryl ester transfer protein (CETP), which transports cholesteryl esters from high-density lipoproteins (HDL) to very low-density lipoproteins (VLDL) and TG [3]. Additionally, thyroid hormones affect lipid regulation from extrahepatic organs, such as adipose tissue, intestines, and muscle [4]. As they affect both efflux and influx of lipid metabolism, physiologically, they are expected to have a crucial role to maintain homeostasis of cholesterol. Although it not clear or determined that subtle changes of thyroid hormone levels within normal ranges could cause significant alterations in lipid profiles, overt changes are expected to cause a deterioration of cholesterol homeostasis.

Significant changes in lipid profiles have been reported in overt thyroid dysfunction [5,6,7,8,9,10]. Although changes in TG, VLDL, and HDL levels have been reported in various instances, it is consistently observed that total cholesterol and LDL-cholesterol levels are increased in overt hypothyroidism and decreased in hyperthyroidism [11,12]. Alterations in lipid profiles in the presence of subtle changes such as subclinical thyroid dysfunction have also been widely observed. In contrast to findings in overt hypothyroidism, no conclusive agreements have been made in lipid profile components in subclinical hypothyroidism, which is characterized by elevated levels of thyrotropin (TSH) and thyroid hormone in the normal range [13,14]. In some studies, total cholesterol and LDL-cholesterol, TG, and HDL-cholesterol levels in subclinical hypothyroidism subjects were not different from those in euthyroid subjects [15,16,17]. Conversely, other research has reported significantly higher levels of total cholesterol, LDL-cholesterol, and TG were found in patients with subclinical hypothyroidism than in euthyroid subjects [18]. In patients with endogenous and exogenous subclinical hyperthyroidism, a decrease in HDL-cholesterol was observed [11].

In addition, only a few studies have evaluated the association between thyroid function and serum lipid profile components in subjects with normal thyroid function. Roos et al. reported a negative correlation between free T4 and total cholesterol and LDL-cholesterol in euthyroid subjects [19]. De Jesus Garduno-Garci et al. reported a positive correlation between free T4 and HDL-cholesterol and between TSH and total cholesterol and TG in both euthyroid subjects and subjects with subclinical hypothyroidism [20]. In contrast, Mehran et al. recently demonstrated a positive correlation between TSH and LDL-cholesterol and TG levels as well as a negative correlation between free thyroxine (T4) and LDL-cholesterol among subjects with normal thyroid function [21]. Currently, there is a lack of consistent results among population-based studies in terms of a correlation between thyroid dysfunction and serum lipid profiles.

In this study, we aimed to study the association between free T4 within reference ranges and serum lipid profiles in the Korean population using the most recent nationally representative data.

## 2. Materials and Methods

### 2.1. Study Subjects

This study was based on the survey data from the sixth Korea National Health and Nutrition Examination Survey (KNHANES VI), a cross-sectional survey conducted by the Korean Centers for Disease Control and Prevention (KCDC) between 2013 and 2015. KNHANES performs assessments by trained interviewers, monitors the health and nutritional status of the Korean population, and notes the trends in the prevalence of diseases during the survey year. The survey consists of a health interview, health examination and nutritional status. The survey uses stratified, multistage clustered probability sampling. KNHANES was approved by the Institutional Review Board of the KCDC. Written informed consent was acquired from all participants since 1998 [22]. This study was approved by Eunpyeong St, Mary’s Hospital Institutional Review Board at the Catholic Medical Center, the Catholic University of Korea (IRB approval No. PC19ZESI0024). Written informed consent was exempted. A total of 8577 subjects were recruited in KNHANES VI.

For our study, 3952 subjects were selected who were older than 19 years and had undergone thyroid function tests and lipid profile testing. Subjects were excluded based on the following criteria: (1) previous therapy for thyroid disease (*n* = 19), (2) abnormal free T4 levels (<0.82 ng/mL or >1.76 ng/mL) (*n* = 70), (3) abnormal TSH levels (<0.62 mIU/L or >6.68 mIU/L) (*n* = 284), (4) current statin therapy (*n* = 15), and (5) treatment for cardiovascular disease (*n* = 16). A total of 3548 subjects were finally selected.

### 2.2. Laboratory Analyses

Blood and urine samples were collected from all participants. Serum free T4, TSH, and thyroid peroxidase antibodies (TPOAb) were analyzed with electrochemiluminescence immunoassay. Free T4, TSH, and TPOAb levels were measured using an E- free T4 kit (Cobas e 411, Roche Diagnositics GmbH Mannheim, Germany), E-TSH kit (Cobas e 801, Roche Diagnostics GmbH, Mannheim, Germany), and E-Anti-TPO kit (Cobas e 801, Roche Diagnostics GmbH, Mannheim, Germany). The reference ranges for free T4 and TPOAb were 0.89 to 1.76 ng/mL and 0–34 IU/mL, respectively. As TSH is affected by iodine intake status, and excess iodine is prevalent in Korea [23], serum TSH levels between 0.62 and 6.68 mIU/L, based on population data [24], were considered as reference ranges. Urine iodine concentration was measured by inductively coupled plasma mass spectroscopy (ICP-MS: PerkinElmer ICP-MS, Waltham, USA) and was adjusted for creatinine concentration.

Serum total cholesterol, TG, and HDL-cholesterol were measured by enzymatic methods using a commercial kit (Sekisui, Osaka, Japan) with a Hitachi Automatic Analyzer 7600 (Hitachi, Tokyo, Japan). LDL-cholesterol was calculated using the Friedewald formula: LDL-cholesterol = total cholesterol − HDL-cholesterol − (TG/5) [25]. Non-HDL-cholesterol was calculated by subtracting the quantity of HDL-cholesterol from total cholesterol [26].

### 2.3. Definition of Other Variables

Body mass index (BMI) was calculated as weight in kilograms divided by the square of height in meters (kg/m^2^). Smoking status was defined as never or current smoker. Heavy alcohol consumption was defined as ≥ 210 g alcohol/week for males and ≥ 140 g alcohol/week for females. Physical activity was defined as activity consisting of at least 30 min of walking for at least 5 days a week.

In this study, dyslipidemia was defined as hypertriglycemia (TG ≥ 150 mg/dL), high LDL-cholesterol (LDL-cholesterol ≥ 130 mg/dL), and low HDL-cholesterol (HDL-cholesterol ≤ 40 mg/dL for males, ≤ 50 mg/dL for females) according to the Adult Treatment Panel III Criteria [27].

### 2.4. Statistical Analysis

All statistical analyses were performed using with SAS survey procedures version 9.3 (SAS Institute, Cary, NC, USA). Continuous variables were presented as mean with standard deviation and categorical variables were described in percentages. Comparisons of basic clinical characteristics according to free T4 quartiles were conducted by analysis of variance (ANOVA) testing for continuous variables and the Chi-square test for categorical variables. Univariate linear regression model was used for the association between free T4 quartiles and lipid profiles and multivariate linear regression was performed to control confounding factors such as age, sex, BMI, smoking status, alcohol consumption, urine iodine, TPOAb, and diabetes mellitus. The association was presented as odds ratios (OR) with 95% confidence intervals. We calculated *p* values for the trends for the associations in each group by logistic regression analysis inclusive of a trend test. *p*-value < 0.05 was considered statistically significant.

## 3. Results

### 3.1. Baseline Clinical Characteristics of the Subjects in Different Free T4 Groups by Population-Based Reference Range

Comparisons of clinical characteristics between subjects within the free T4 quartiles are presented in Table 1. Subjects in the highest free T4 quartile were younger than subjects in the other free T4 quartiles. Significant male predominance was observed in the higher quartiles (51.2% in Q3 and 70.9% in Q4) compared with the lower quartiles (43.9 % in Q1 and 52.4% in Q2, *p* < 0.001). Habitual factors including alcohol consumption and physical activity were not significantly different between the four groups. However, smoking prevalence was higher in the upper quartiles than that in the lower quartiles (*p* < 0.001). The highest total cholesterol levels were observed in the lowest free T4 quartile. Free T4 levels were negatively correlated with total serum cholesterol levels. Other lipid profiles such as TG, HDL-cholesterol, and LDL-cholesterol were not significantly different between the free T4 quartiles. A significant decrease in serum non-HDL-cholesterol levels was observed in the higher free T4 quartiles (*p* = 0.001).

### 3.2. Changes in Lipid Profiles According to Free T4 Quartiles

Total serum cholesterol, LDL-cholesterol and non-HDL cholesterol decreased significantly with higher free T4 quartiles (*p* for trend <0.001, 0.013, and <0.001, respectively). However, after adjusting for age, sex, BMI, behavioral patterns, urine iodine, TPOAb positivity, presence of diabetes mellitus, and TSH, the statistical significance disappeared. In contrast, a significant decrease in TG levels and an increase in HDL-cholesterol levels was observed in higher free T4 quartiles after adjusting for possible confounding factors (Table 2).

### 3.3. Risk of Dyslipidemia According to Free T4 Quartiles

Although higher free T4 quartiles were associated with a decreased risk of dyslipidemia in a crude model, statistical significance was not maintained after adjustment for confounding factors (Table 3). However, the risk of hypertriglyceridemia decreased significantly with the increase in free T4 quartiles after adjustment for confounders (*p* for trend = 0.002, 0.026, 0.043, 0.022, and 0.035 for models 1–5, respectively). Compared with the lowest free T4 quartile, the highest free T4 quartile showed a reduced risk of hypertriglyceridemia with an OR of 0.72 (95% confidential interval 0.53–0.98) after adjustment for all confounding factors. A decrease in the risk of high LDL-cholesterol was observed with higher free T4 quartiles (*p* for trend = 0.04). However, significance disappeared after adjustment for confounders. A significant decrease in non-HDL-cholesterol in higher free T4 quartiles was observed only in model 2 (adjustment for age and sex, *p* for trend = 0.019).

## 4. Discussion

In the present study, differences in lipid profiles were evaluated according to free T4 quartiles between subjects with normal thyroid function. Higher free T4 quartiles were associated with a significant decrease in serum TG levels while it was associated with a significant increase in serum high-density lipoprotein (HDL) cholesterol after adjustment for confounding factors. However, there were no significant changes in other lipid profiles including total cholesterol, LDL-cholesterol, and non-HDL-cholesterol based on free T4 quartiles. Regarding the presence of dyslipidemia, the risk of hypertriglyceridemia (≥ 150 mg/dL) significantly decreased as free T4 quartiles increased. Although some significant correlations were found, no consistent results were observed in the analysis of free T4 quartiles and other parameters of dyslipidemia.

In lipid metabolism, thyroid hormone has significant effects on both influx and efflux of lipid mainly by its action on hepatic lipid processes [1]. Specifically, it facilitates hepatic free fatty acid uptake by upregulation of fatty acid transporting proteins such as fatty acid transporter protein, liver fatty acid binding protein, and fatty acid translocase [28]. De novo lipogenesis, which converts excessive glucose to fatty acid, is also regulated directly and indirectly by thyroid hormone. As increased hepatic fatty acids are related to subsequent triacylglycerol assembly and VLDL production, actions of thyroid hormone are crucial for lipid influx [4]. Conversely, thyroid hormone is involved in the degradation of fatty acids, efflux of lipid, by activation of several lipases, lyphophagy, peroxisomal fat oxidation and mitochondrial oxidation [29]. Also, thyroid hormone induces cholesterol biosynthesis by activating hydroxymethylglutaryl coenzyme A reductase, a rate-limiting enzyme, and increases the expression of LDL-cholesterol receptors on the cell surface [10]. Additionally, CETP, which transports cholesteryl esters from HDL-cholesterol to LDL-cholesterol and VLDL, is regulated by thyroid hormone [30]. Thyroid hormone enhances the activity of lipoprotein lipase, which facilitates the hydrolysis of TG containing proteins and the transfer of cholesterol from lipoprotein to HDL-cholesterol [12]. As it controls cholesterol excretion by upregulation of bile acid synthesis and excretion [31], thyroid hormone plays a vital role in cholesterol metabolism in general (from synthesis to clearance). Furthermore, thyroid hormone regulates lipid metabolism of extrahepatic organs such as adipose tissue, intestines and muscle [4]. Thyroid hormone induces lipolysis in adipose tissue by an enhanced catecholamine-mediated mechanism [32]. Excretion of bile acids in intestines, a process related to cholesterol clearance, is promoted by thyroid hormone. Thyroid hormone activates free fatty acid uptake in muscle by increased local lipoprotein lipase activity [1,33]. As thyroid hormone is involved in important processes of lipid influx and efflux, its net effects on lipid profiles are difficult to be estimated especially in cases of normal thyroid function. Based on our results, it could be assumed that free T4 within normal range mainly exerts a positive influence on cholesterol metabolism in terms of TG and HDL cholesterol.

A substantial number of studies demonstrated hypertriglyceridemia and higher levels of LDL-cholesterol in overt hypothyroidism compared with euthyroid subjects [3,12,34]. The effect of overt hypothyroidism on HDL-cholesterol has been controversial [12,14]. In contrast to overt hypothyroidism, there have been conflicting data regarding the relationship between subclinical hypothyroidism and lipid profile alterations. In a small-scale study, significant alternations were not observed in lipid profiles of subjects with subclinical hypothyroidism when compared to euthyroid subjects [34]. By contrast, the Colorado study of 25862 subjects with TSH levels in the reference range suggested that a significant positive correlation exists between total cholesterol, TG, and LDL-cholesterol and increasing TSH [18]. However, these studies did not adjust for age, sex, and BMI, which could affect lipid profiles. A study from the National Health and Nutrition Examination Survey III showed no significant alterations in lipid profiles after adjustment for age, race, sex, and statin use in a subclinical hypothyroid group [15]. In a community-based study involving 2108 participants in Australia, although total cholesterol, TG, and LDL-cholesterol increased with TSH levels, no significant differences were seen in these lipid profiles according to TSH levels after adjustment for age and sex [19]. In contrast to these two studies, the Tromsø study demonstrated a positive correlation between total cholesterol and LDL-cholesterol with TSH levels after adjustment for age, BMI, and smoking status [35]. Various sample size and varying methods for adjustment of confounding factors such as age and other metabolic components could explain the conflicting results from previous studies.

There have been several research projects about an association between TSH and lipid profiles in euthyroidism [19,20,21,35,36,37,38,39,40]. Although there have been several studies in euthyroid subjects as mentioned, there have only been a few studies evaluating the association between thyroid hormone (free T4) and lipid profiles. One cross-sectional study in the Netherlands suggested that normal range free T4 showed a negative correlation with total cholesterol, TG, and LDL-cholesterol after adjustment for age and sex, but a positive correlation with HDL-cholesterol in euthyroid subjects (defined as 0.35 ≤ TSH ≤ 4.94 mIU/L) [18]. However, TSH showed a significant positive correlation with TG only. Another multi-center study evaluated the correlation of lipid profiles with both TSH and free T4 levels in subclinical thyroidisim (defined as 4.5 ≤ TSH ≤ 10.0 mIU/L) and euthyroid subjects (0.25 ≤ TSH ≤ 4.5 mIU/L) [20]. While serum TSH level showed a positive correlation with total cholesterol and TG, free T4 showed a positive correlation with HDL-cholesterol and a negative correlation with TG in the study. The significant correlation between free T4 and LDL-cholesterol disappeared after adjustment for possible confounding factors such as age and sex. In contrast to a previous study [19], de Jesus Garduno-Garci et al [20] suggested that TSH is a more suitable indicator of LDL-cholesterol and TG than free T4. Another study based on data from a prospective and large sample size population reported a correlation between thyroid dysfunction and metabolic marker in subclinical hypothyroidism (TSH <10 mIU/L) and euthyroid (0.3 ≤ TSH ≤ 5.06 mIU/L) non-obesity subjects [21]. TSH was positively correlated with total cholesterol, LDL-cholesterol, and TG, while no significant association between TSH and HDL-cholesterol was found. In contrast to the study by de Jesus Garduno-Garci et al. [20], free T4 showed a negative correlation with total cholesterol and LDL-cholesterol and there was no significant difference in TG after adjustment for confounders such age, sex, BMI, and the homeostasis model assessment index for insulin resistance. However, populations used in those studies would not be considered as national representative data. Our study, utilized national representative subjects with normal thyroid function, suggested the association of free T4 with serum triglyceride and HDL-cholesterol levels. In fact, an association of TSH and lipid profiles was also evaluated in our data (Appendix A and Appendix A) as several previous studies had analyzed. However, compared to analysis of free T4 and lipid profiles, a statistical significance in relationship between TSH and lipid profiles was not definite according to confounding factors.

Recently, a study using KNHANES IV data investigated the association between thyroid dysfunction and lipid profiles according to age and sex [40]. Although the study reported free T4 as an independent variable for reducing the risk of hypertriglyceridemia, it was analyzed in all subjects, including dose with overt hyperthyroidism and hypothyroidism, who had undergone the thyroid function test. Additionally, in contrast to our study, Oh et. al. [40] did not exclude subjects with taking lipid-lowering agents.

There are some limitations to this study. First, as this study was based on a cross-sectional survey, a causal relationship could be not determined. Second, because laboratory tests were performed only once, inter-individual variations could not be considered. Another limitation is that we could not estimate lipid profiles according to thyronine T3, which is a biologically active form of thyroid hormone. Data on T3 was not acquired in KNHANES VI.

This study has several strengths. First, although there were several studies elucidating the association between thyroid function and lipid profiles [3,12,19,20,35,36,37,38,39,40], evaluations of the associations of thyroid hormone (free T4) in euthyroid populations were sparse [19,20,39,40], and our study utilized nationwide representative data. In addition, in contrast to several previous studies [36,37,38] which reported significant alterations according to TSH without analyzing free T4, our study evaluated associations between free T4 and lipid profiles, and significant relationships were observed. Second, population-based TSH reference ranges were used for defining normal thyroid function as ethnicity and iodine intake could have influenced TSH levels [41]. Third, subjects with a history of cardiovascular disease who would be expected to be taking lipid lowering agents as well as subjects on medication for dyslipidemia were excluded. Additionally, TPOAb, which is related to higher TSH, was considered as a confounding factor [42].

## 5. Conclusions

Thyroid hormones have multiple effects on lipid metabolism. We demonstrated an association between free T4 levels within normal range and dyslipidemia in a nation-wide data. The results of our study suggest that subtle changes in serum free T4 levels, even in the reference ranges, could be related to alterations in lipid profiles, specifically in triglycerides and HDL-cholesterol. In particular, the lowest free T4 quartiles were related to increased risk of hypertriglycemia which is a dyslipidemia component. Further studies are needed to confirm the relationship between thyroid hormones including T3, a biologically active form of thyroid hormone, and lipid profiles. Given our results, cautious monitoring of subjects with normal but relatively low free T4 levels would be required for management of dyslipidemia.

## Figures and Tables

**Table 1 jcm-08-00758-t001:** Baseline characteristics of the patients based on free T4 levels.

Clinical Parameters	Q1 (free T4 <1.13 ng/mL)	Q2 (1.13–1.24 ng/mL)	Q3 (1.25–1.34 ng/mL)	Q4 (free T4 >1.35 ng/mL)	*p* Value
Patients (*n*, %)	877 (24.3)	906 (25.1)	834 (23.4)	931 (27.2)	
Age (years)	40.1 ± 0.5	44.9 ± 0.5	42.2 ± 0.6	38.5 ± 0.5	<0.001
Sex (male *n*, %)	295 (36.1)	401 (47.6)	405 (51.2)	635 (70.9)	<0.001
BMI (Kg/m^2^)	23.9 ± 0.1	24.0 ± 0.1	23.61 ± 0.1	23.6 ± 0.1	0.047
Smoking (*n*, %)	134 (16.1)	186 (20.4)	192 (24.4)	258 (28.1)	<0.001
Heavy alcohol consumption (*n*, %)	99 (11.4)	99 (11.5)	108 (12.8)	125 (14.0)	0.395
Physical activity (*n*, %)	144 (16.5)	141 (14.8)	145(17.7)	169 (17.4)	0.460
Diabetes mellitus (*n*, %)	36 (3.6)	31 (3.3)	35 (4.9)	27 (2.9)	0.258
Urine iodine: (mcg/g)					<0.001
Q1 (<86.20)	176 (20.0)	201 (22.7)	204(23.7)	303(32.7)	
Q2 (86.2–165.4)	170 (18.5)	223 (23.7)	234 (28.9)	260 (28.4)	
Q3 (165.5–404.3)	240 (28.5)	228 (25.7)	207 (26.1)	201 (20.4)	
Q4 (>404.3)	291 (32.9)	254 (27.9)	189 (21.4)	167 (18.4)	
Total cholesterol (mg/dL)	193.9 ± 1.6	192.4 ± 1.3	189.2 ± 1.5	187.2 ± 1.2	0.001
Triglyceride (mg/dL)	144.4 ± 6.0	145.4 ± 5.0	138.2 ± 4.9	129.4 ± 4.1	0.056
HDL-cholesterol (mg/dL)	51.0 ± 0.5	50.3 ± 0.4	51.5 ± 0.5	51.2 ± 0.5	0.328
LDL-cholesterol (mg/dL)	114.0 ± 1.5	113.0 ± 1.3	110.1 ± 1.2	110.1 ± 1.2	0.074
non- HDL-cholesterol (mg/dL)	142.9 ± 1.6	142.1 ± 1.2	137.8 ± 1.5	136.0 ± 1.2	<0.001

Data are expressed as mean ± standard error or number including percentage; Free T4: free thyroxine; BMI: body mass index; HDL: high-density lipoprotein; LDL: low-density lipoprotein.

**Table 2 jcm-08-00758-t002:** Changes in lipid profiles according to free T4 quartiles.

Lipid Profiles	Q1 (free T4 <1.13 ng/mL)	Q2 (1.13–1.24 ng/mL)	Q3 (1.25–1.34 ng/mL)	Q4 (free T4 >1.35 ng/mL)	*p* for Trend
**Total cholesterol** (mg/dL)					
Crude	Reference (1.0)	−1.6 ± 2.0	−4.7 ± 2.1	− 6.8 ± 1.9	<0.001
Model 1	Reference (1.0)	−0.4 ± 2.0	−2.2 ± 2.1	−2.3 ± 1.9	0.143
Model 2	Reference (1.0)	−0.4 ± 2.0	−1.5 ± 2.0	−1.4 ± 1.9	0.374
Model 3	Reference (1.0)	−0.4 ± 2.0	−1.6 ± 2.0	−1.3 ± 1.8	0.378
Model 4	Reference (1.0)	−0.3 ± 2.0	−1.2 ± 2.0	−1.1 ± 1.9	0.488
Model 5	Reference (1.0)	0.1 ± 2.0	− 0.7± 2.1	−0.5 ± 1.9	0.395
**Triglyceride** (mg/dL)					
Crude	Reference (1.0)	1.0 ± 7.5	−6.2 ± 8.0	−15.1 ± 1.3	0.056
Model 1	Reference (1.0)	−5.1 ± 7.1	−12.5 ± 7.8	−32.0 ± 7.8	<0.001
Model 2	Reference (1.0)	−5.3 ± 7.0	−9.6 ± 7.7	−27.51 ± 7.7	0.001
Model 3	Reference (1.0)	−4.7 ± 6.9	−9.8 ± 7.6	−26.0 ± 7.6	0.001
Model 4	Reference (1.0)	−4.9 ± 6.9	−11.3 ± 7.5	−26.9 ± 7.7	0.001
Model 5	Reference (1.0)	−4.8 ± 7.1	−11.1 ± 7.7	−26.8 ± 8.0	0.001
**HDL-cholesterol** (mg/dL)					
Crude	Reference (1.0)	−0.7 ± 0.7	0.44 ± 0.7	0.2 ± 0.6	0.412
Model 1	Reference (1.0)	−0.1 ± 0.6	1.0 ± 0.7	1.74 ± 0.6	0.002
Model 2	Reference (1.0)	−0.2 ± 0.6	0.6 ± 0.6	1.2 ± 0.6	0.036
Model 3	Reference (1.0)	−0.1 ± 0.6	0.6 ± 0.6	1.2 ± 0.6	0.026
Model 4	Reference (1.0)	−0.1 ± 0.6	0.7 ± 0.6	1.3 ± 0.6	0.014
Model 5	Reference (1.0)	0.1 ± 7.6	0.7 ± 0.6	1.3 ± 0.6	0.014
**LDL-cholesterol** (mg/dL)					
Crude	Reference (1.0)	−1.0 ± 1.9	−3.9 ± 2.0	−3.9 ± 1.2	0.013
Model 1	Reference (1.0)	0.7 ± 1.9	−0.6 ± 2.0	2.3 ± 1.9	0.337
Model 2	Reference (1.0)	0.8 ± 1.9	−0.1 ± 1.9	3.0 ± 1.8	0.167
Model 3	Reference (1.0)	0.6 ± 1.8	−0.3 ± 1.9	2.7 ± 1.8	0.213
Model 4	Reference (1.0)	0.8 ± 1.8	0.4 ± 1.9	3.0 ± 1.8	0.129
Model 5	Reference (1.0)	1.2 ± 1.8	0.8 ± 1.9	3.5 ± 1.9	0.080
**Non-HDL-cholesterol** (mg/dL)					
Crude	Reference (1.0)	−0.8 ± 1.9	−5.2 ± 2.1	−6.9 ± 1.9	<0.001
Model 1	Reference (1.0)	−03 ± 1.9	−3.1 ± 2.1	−4.1 ± 1.9	0.013
Model 2	Reference (1.0)	−0.2 ± 1.9	−2.1 ± 2.0	−2.5 ± 1.9	0.105
Model 3	Reference (1.0)	−0.3 ± 1.9	−2.3 ± 2.0	−2.6 ± 1.8	0.096
Model 4	Reference (1.0)	−0.2 ± 1.9	−1.9 ± 2.0	−2.4 ± 1.9	0.133
Model 5	Reference (1.0)	0.2 ± 1.9	−1.4 ± 2.0	−1.8 ± 1.9	0.233

The association between free T4 quartiles and lipid profiles were performed by linear regression. Model 1: adjusted for age and sex; Model 2: Model 1 + body mass index; Model 3: Model 2 + smoking, alcohol consumption, and physical activity; Model 4: Model 3 + urine iodine, peroxidase antibody, and diabetes mellitus; Model 5: Model 4 + thyrotropin.

**Table 3 jcm-08-00758-t003:** Risk of dyslipidemia according to free T4 quartiles.

Presence of Dyslipidemia or Its Component	Q1 (Free T4 <1.13 ng/mL)	Q2 (1.13–1.24 ng/mL)	Q3 (1.25–1.34 ng/mL)	Q4 (Free T4 >1.35 ng/mL)	*p* for Trend
**Dyslipidemia**					
Crude	Reference (1.0)	0.90(0.73–1.10)	0.77 (0.61–0.97)	0.65 (0.53–0.80)	<0.001
Model 1	Reference (1.0)	0.92 (0.75–1.13)	0.83 (0.66–1.06)	0.74 (0.59–0.92)	0.006
Model 2	Reference (1.0)	0.92 (0.73–1.15)	0.88 (0.68–1.14)	0.80 (0.63–1.01)	0.072
Model 3	Reference (1.0)	0.92 (0.74–1.15)	0.87 (0.68–1.13)	0.81 (0.64–1.03)	0.084
Model 4	Reference (1.0)	0.92 (0.74–1.15)	0.86 (0.67–1.11)	0.80 (0.63–1.01)	0.060
Model 5	Reference (1.0)	0.92 (0.74–1.15)	0.86 (0.67–1.12)	0.81 (0.64.1.02)	0.069
**Hypertriglyceridemia**					
Crude	Reference (1.0)	0.99 (0.78–1.26)	0.97 (0.75–1.25)	0.83 (0.65–1.06)	0.125
Model 1	Reference (1.0)	0.90 (0.70–1.17)	0.89 (0.67–1.17)	0.65 (0.50–0.86)	0.002
Model 2	Reference (1.0)	0.90 (0.68–1.18)	0.95 (0.71–1.27)	0.71 (0.53–0.95)	0.031
Model 3	Reference (1.0)	0.90 (0.68–1.19)	0.95 (0.71–1.27)	0.73 (0.54–0.97)	0.047
Model 4	Reference (1.0)	0.89 (0.67–1.17)	0.92 (0.68–1.23)	0.70 (0.52–0.95)	0.027
Model 5	Reference (1.0)	0.91 (0.68–1.20)	0.94 (0.69–1.26)	0.72 (0.53–0.98)	0.044
**High LDL-cholesterol**					
Crude	Reference (1.0)	0.94 (0.75–1.18)	0.74 (0.57–0.97)	0.82 (0.64–1.04)	0.040
Model 1	Reference (1.0)	1.01 (0.80–1.29)	0.87 (0.66–1.15)	1.12 (0.87–1.44)	0.665
Model 2	Reference (1.0)	1.02 (0.80–1.30)	0.90 (0.68–1.18)	1.17 (0.90–1.51)	0.427
Model 3	Reference (1.0)	1.01 (0.79–1.28)	0.88 (0.67–1.16)	1.14 (0.88–1.48)	0.525
Model 4	Reference (1.0)	1.02 (0.80–1.30)	0.91 (0.69–1.20)	1.17 (0.90–1.51)	0.386
Model 5	Reference (1.0)	1.05 (0.82–1.34)	0.94 (0.71–1.23)	1.22 (0.93–1.58)	0.255
**Non-HDL-cholesterol**					
Crude	Reference (1.0)	0.88 (0.70–1.12)	0.76 (0.59–0.98)	0.66 (0.52–0.85)	<0.001
Model 1	Reference (1.0)	0.90 (0.70–1.15)	0.82 (0.64–1.06)	0.75 (0.58–0.97)	0.019
Model 2	Reference (1.0)	0.89 (0.69–1.15)	0.86 (0.66–1.12)	0.80 (0.61–1.04)	0.087
Model 3	Reference (1.0)	0.89 (0.69–1.15)	0.85 (0.66–1.11)	0.80 (0.61–1.05)	0.086
Model 4	Reference (1.0)	0.90 (0.70–1.17)	0.88 (0.68–1.15)	0.82 (0.63–1.07)	0.135
Model 5	Reference (1.0)	0.92 (0.70–1.19)	0.89 (0.68–1.17)	0.84 (0.64–1.10)	0.184

The association between free T4 quartiles and dyslipidemia and component of dyslipidemia were performed by logistic regression analysis considering confounders. Dyslipidemia was defined as hypertriglycemia (TG ≥ 150 mg/dL), high LDL-cholesterol (LDL-cholesterol ≥ 130 mg/dL), and low HDL-cholesterol (HDL-cholesterol ≤ 40 mg/dL for males, ≤ 50 mg/dL for females). Model 1: adjusted for age and sex; Model 2: Model 1 + body mass index; Model 3: Model 2 + smoking, alcohol consumption, and physical activity; Model 4: Model 3 + urine iodine, peroxidase antibody, and diabetes mellitus; Model 5: Model 4 + thyrotropin.

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
