# Peer review of "High Normal Range of Free Thyroxine is Associated with Decreased Triglycerides and with Increased High-Density Lipoprotein Cholesterol Based on Population Representative Data"

_jcm, 2019, doi:10.3390/jcm8060758_

Reviewer 1 Report

In the manuscript, Lee et al, analyzed the association of thyroid hormone (free thyroxine, free T4) with lipid profiles on a population representative data. The study included a large group of study subjects. The study indicated that increased free T4 levels are associated with decreased TG and with increased HDL-C levels. The author considered different confounding factors for the analysis that is an advantage over other previous studies. The conclusion drawn from the study is supported by the survey data and could be beneficial to the clinical outcomes

Few minor corrections.

Line 116, repeated word ‘were used’

Line 131, sentence is incomplete

Line 170, is that decrease or increased in serum HDL level?

Line 269, Inconsistent reference formatting.

Author Response

Dear reviewer,

We  appreciate the efforts of you in the review my manuscript. I have made some corrections and clarifications in the manuscript considering the comments and suggestions.  It is my hope that the enclosed revision will now be judged acceptable for publication in Journal of Clinical Medicine.

Reviewer 2 Report

The authors present and interesting topic here on High Normal Range of Free Thyroxine is Associated with Decreased Triglycerides and with Increased High Density Lipoprotein Cholesterol Based on Population Representative Data. A well conducted research. Issues which will need to be addressed are:

Abstract:

line 22 increased needs to be changed to increase

line 26-27 “Serum triglyceride and HDL cholesterol levels are associated with serum free T4 levels within normal reference ranges” – what are they associated with? Sentence is unclear

Introduction:

Line 42 please consider adding the reference Effects of Thyroid Dysfunction on Lipid Profile (https://www.ncbi.nlm.nih.gov/pmc/articles/PMC3109527/)  at the end of the sentence as it explains about the various lipid abnormalities in endocrinologic disorders.

In methods section please explain in confounders to the study and how they were resolved.

Results look ok

Discussion section: Discuss how thyroid hormone is involved in the lipid metabolism

In conclusion: please expand the paragraph, looks too short

English and grammar need to be thoroughly checked throughout the manuscript

Author Response

Dear reviewer,

We appreciate your suggestion and agree with your opinion. Your comments were very helpful to improve the quality of revised manuscript. A point-by-point response to the reviewer was made and manuscript was revised accordingly. 

Reviewer 3 Report

The manuscript explored the association of free thyroxine levels and lipid profiles in euthyroid subjects in a cross-sectional study based on national survey data. The study is interesting and reports results that would potentially provide an important contribution to clinical research, considering that dyslipidemia constitutes a major risk factor for premature atherosclerosis and cardiovascular disease. Though the manuscript is generally well written, it needs some improvements.

Introduction reports results from some previous studies that are subsequently discussed, while it should briefly describe the physiological and molecular mechanisms underlying the relationship between thyroid function and lipid metabolism, and the consequences to health. On the other hand, in discussion the authors mention some concepts already presented but not clarified in the introduction, making the reading a bit difficult.

In results, the authors should refer to the statistical methods used below tables 2 and 3.

Moreover, did the authors perform also analyses on the association between TSH increasing quartiles and the lipid profiles even though they did not show data? it would be interesting to compare these results with those of previous investigations.

Minor revisions:

I suggest a revision of the text from an English-mother tongue.

At line 116, the authors wrote “were used” twice.

Author Response

Dear reviewer, 

We appreciate your efforts for reviewing our manuscript. Your comment was very helpful to improve the quality of revised manuscript. A point-by-point response to the referees was made and manuscript was revised accordingly. 

Reviewer 4 Report

The manuscript “High normal range of free thyroxine is associated with decreased triglycerides and with increased high density lipoprotein cholesterol based on population representative data”, by Lee and colleagues, is a study that, based on Korean cross-sectional survey data, identifies an association between serum triglyceride and HDL cholesterol levels with serum free T4 levels within normal reference ranges.

The study appears well conducted and described, despite the lack of intra-individual information and T3 measurements.

Minor points:

Can raw data be provided as supplementary material (or on request) to the scientific community?

- line 22: increased or increase?

- line 23: confound or confounding?

- line 52: founded or found?

- line 89: could the E- (?) kit (or kits) be more specifically named (including commercial code)?

- line 96: could the enzymatic methods be more specifically mentioned (including commercial code)?

- line 104: waking or walking?

- line 116: “were used” is repeated two times

- line 131: were or was?

- line 191: there is an extra comma

- line 196: thyroidisim or thyroidism?

- line 211: euthyroid or euthyroidism?

Table titles: Although already mentioned in the Materials and Methods section, the unit of measure for T4 levels, mentioned in the quartiles, could be added in the table titles.

Author Response

Dear reviewer,

We appreciate your efforts for reviewing our manuscript. Your comment was very helpful to improve the quality of revised manuscript. A point-by-point response to the referees was made and manuscript was revised accordingly. 

Round  2

Reviewer 3 Report

The authors have responded to all questions and significantly improved the manuscript. If the editor agrees, I would propose, in agreement with the authors, to publish the results on the association between TSH increasing quartiles and lipid profiles as supplementary material.

I point out only minor revisions:

Line 25: “with” should be deleted;

Line 41: “effect” or “affect”?

Line 64: “association between free T4 AND HDL-cholesterol….”

Line 65: “euthyroid subjects” and not “euthyroid subject”

Line 79: please eliminate “and”

Line 128: “and” should be eliminated

Line 226: consider to remove “which”

Line 232: do you mean “hypothyroidism”?

Lines 228, 229, 244: consider to use other terms beyond “inconsistent” such as “conflicting”, “controversial”, “fragmentary”, etc.

Line 252: please correct “Another”

Line 279: please space “acquired” and “in”

Line 300: please correct typo in “results”

Author Response

Dear an editor and a reviewer,

We appreciate your and reviewer's efforts for reviewing our manuscript. In the revised

manuscript, we made corrections according to reviewer's comments and suggestions. A

response to the referee's suggestions has been listed one by one. Modified contents were

written in highlight in the manuscript and we used the "Track Changes" function in Microsoft

Word. We made a correction for some typos. We deleted the incorrectly used conjunction and relative pronoun.
